# A Logic for Expressing Log-Precision Transformers

**William Merrill**
New York University
willm@nyu.edu

**Ashish Sabharwal**
Allen Institute for AI
ashishs@allenai.org

## Abstract

One way to interpret the reasoning power of transformer-based language models is to describe the types of logical rules they can resolve over some input text. Recently, Chiang et al. (2023) showed that finite-precision transformer classifiers can be equivalently expressed in a generalization of first-order logic. However, finite-precision transformers are a weak transformer variant because, as we show, a single head can only attend to a constant number of tokens and, in particular, cannot represent uniform attention. Since attending broadly is a core capability for transformers, we ask whether a minimally more expressive model that can attend universally can also be characterized in logic. To this end, we analyze transformers whose forward pass is computed in $\log n$ precision on contexts of length $n$. We prove any log-precision transformer classifier can be equivalently expressed as a first-order logic sentence that, in addition to standard universal and existential quantifiers, may also contain majority-vote quantifiers. This is the tightest known upper bound and first logical characterization of log-precision transformers.

---

Any log-precision transformer can be re-expressed as a sentence in $\mathsf{FO}(\mathsf{M})$ logic, e.g.:

$$\mathsf{M}i.\, \mathtt{a}(i)\ \wedge\ \mathsf{M}j.\, \mathtt{b}(j)\ \wedge\ \neg\exists k, \ell.\, (\mathtt{a}(k) \wedge \mathtt{b}(\ell) \wedge \ell < k)$$

*(m a's followed by m b's, i.e., $a^m b^m$)*

aaaabbbb ✓     aaabbbbb ✗     baaaabbb ✗

---

Figure 1: A first-order logic with majority ($\mathsf{FO}(\mathsf{M})$) sentence for $\mathtt{a}^m\mathtt{b}^m$. In addition to standard $\forall$ and $\exists$ quantifiers over string indices, $\mathsf{FO}(\mathsf{M})$ allows *majority* quantifiers ($\mathsf{M}$) that take a majority-vote across indices. $\mathtt{a}(i)$ indicates whether token $i$ is $\mathtt{a}$ (and analogously for $\mathtt{b}$). We prove $\mathsf{FO}(\mathsf{M})$ can express any function computed by a log-precision transformer.

## 1 Introduction

The incredible success of deep learning models, especially very large language and vision transformers with hundreds of billions of parameters (Brown et al., 2020; Thoppilan et al., 2022), has come at the cost of increasingly limited understanding of how these models actually work and when they might fail. This raises many concerns, such as around their safe deployment, fairness, and accountability. Does the inner working of a transformer defy description in a simpler symbolic system that we can better understand? Or *can transformer computation be described using a familiar symbolic formalism?* Understanding how to view the reasoning process of a transformer in terms of logic could potentially expand our ability to formally reason about their behavior over large domains of inputs.

Chiang et al. (2023) provide a partial answer to this question, showing that any *finite-precision* transformer classifier can be expressed as a sentence in a variant of first-order logic with counting

quantifiers and modular arithmetic over input position indices. Specifically, counting quantifiers take the form $\exists^{=x} i : \phi(i)$ where $x$ is a count variable and $i$ is a position index. They show that there exists a single sentence in this logic that computes the output of the transformer for any input string of any length. This is a powerful result because it shows that a simple logical formalism is fully sufficient to describe all the complexity of a massive finite-precision transformer. It also provides an upper bound on finite-precision transformers: any function that cannot be defined in first-order counting logic with modular indexing cannot be expressed by the transformer.

However, Chiang et al.'s result is not fully general because it relies on the transformer precision being fixed with respect to the transformer's context length. More generally, as we will demonstrate in Section 3, finite-precision transformers are a fundamentally weak variant of transformers: crucially, cannot express uniform attention patterns, which are a core algorithmic primitive of transformers (Weiss et al., 2018). In fact, we show that they can only attend to a constant number of input positions, which may be seen as a rather limited generalization of hard attention.[1] For example, Chiang et al. show that their logic for finite-precision transformers cannot recognize $\texttt{a}^m\texttt{b}^m$, whereas in practice, transformers can (Bhattamishra et al., 2020).[2] This motivates studying a formal model of transformers where precision grows with context length (which we formalize as *log-precision*), making it possible to capture uniform attention as well as other broad attention patterns. This is useful both for recognizing $\texttt{a}^m\texttt{b}^m$ and more generally for reasoning globally over the input.

We demonstrate that *log-precision* transformer classifiers can also be expressed as sentences in a simple logic: *first-order logic with majority*, or FO(M), over inputs strings (Barrington et al., 1990). In addition to standard existential and universal quantifiers, FO(M) has *majority* quantifiers that return true iff more than half the propositions they quantify are true. It also allows comparing input positions (e.g., $\ell < k$ in Figure 1) and accessing their individual bits. Our main result is as follows:

**Theorem 1** (Informal version of Theorem 2). *For any log-precision transformer $\mathcal{T}$, there exists an* FO(M) *sentence $\phi$ that computes the same function as $\mathcal{T}$, i.e., $\phi(x) = \mathcal{T}(x)$ for any input string $x$.*

**Upper bound.** Theorem 2 shows transformers with more than finite precision can also be expressed in a simple extension of first-order logic, going beyond Chiang et al. (2023)'s result. On the other hand, FO(M) is a strict superset of Chiang et al.'s counting logic; it can simulate counting quantifiers (see Section 2.2) and allows non-modular position comparisons. Thus, handling a more general class of transformers powerful enough to express uniform attention slightly weakens the bound.

Still, our result constitutes (to our knowledge) the tightest upper bound on log-precision transformers and the first defined in terms of logic, building on a line of complexity-theoretic work analyzing the power of transformers (Hahn, 2020; Merrill et al., 2022; Liu et al., 2023; Merrill & Sabharwal, 2023). In particular, FO(M) strengthens the upper bound of log-space-uniform $\mathsf{TC}^0$ by Merrill & Sabharwal (2023). The refined bound adds to the limitations of transformers identified by Merrill & Sabharwal (2023): for example, it establishes unconditionally that log-precision transformers cannot compute boolean matrix permanents, and shows that, in a certain formal sense, integer division and matching parentheses are among the formally hardest problems that transformers can solve (see Section 4).[3]

**Mechanistic interpretability.** Beyond providing an upper bound on the reasoning problems solvable by transformers, we believe Theorem 1 could guide the design of "transformer-complete" programming languages similar in spirit to RASP (Weiss et al., 2018). RASP is a declarative programming language designed to capture transformer computation, and Lindner et al. (2023) implement a compiler from RASP *into* transformers. Unlike RASP, FO(M) can provably express any transformer (Theorem 1), which we believe justifies using it (or an equivalent but more user-friendly variant) as a target language for programs extracted *from* transformers.

Similar to a decision tree, an FO(M) sentence has the interpretable property that each sub-sentence corresponds to a constraint on input (see Figure 1). In contrast, the internal modules of a transformer or circuit do not satisfy this since they map between arbitrary latent spaces. We speculate this property

---

[1]Hard attention is provably substantially weaker than general attention (Hao et al., 2022; Merrill et al., 2022).

[2]Technically, the empirical results of Bhattamishra et al. (2020) are for $\texttt{a}^m\texttt{b}^m\texttt{c}^m$, a harder variant of $\texttt{a}^m\texttt{b}^m$.

[3]To be clear, Theorem 1 is one-sided: every transformer can be expressed as an FO(M) sentence, but not necessarily the other way. Moreover, we believe that many FO(M) sentences *cannot* be expressed by transformers. An exact logical characterization of transformers remains an open problem.

could facilitate interpreting models by translating them to $\mathsf{FO}(\mathsf{M})$, though a careful exploration of the algorithmic and HCI aspects of this idea lies outside the current paper's theoretical scope.

**Contributions.** Our results shed new light on how to view the computation inside transformers in terms of logic. Specifically, our main contributions are to prove the following:

1. Fixed-precision transformers can only attend to a fixed number of tokens, and those with precision less than $\log \log n$ cannot uniformly attend over length-$n$ contexts (Proposition 1).
2. Log-precision transformer classifiers can be expressed as sentences in $\mathsf{FO}(\mathsf{M})$ (Theorem 2).

## 2 Preliminaries: Transformers and $\mathsf{FO}(\mathsf{M})$

Let $\Sigma$ be a finite alphabet. We denote by $^*$ the Kleene star operator, i.e., for a set $X$, $X^* = \bigcup_{n=0}^{\infty} X^n$. We will view transformers and $\mathsf{FO}(\mathsf{M})$ sentences both as functions from $\Sigma^* \to \{0, 1\}$, and show that any function a transformer computes can also be computed by an $\mathsf{FO}(\mathsf{M})$ sentence.

### 2.1 Transformers

We view the transformer precision $p$ as a function of the context length $n$, writing $p(n)$ where appropriate. Let $\mathbb{D}_p$ be the datatype of $p$-precision floats, i.e., tuples $\langle m, e \rangle$ where $m, e$ are signed integers together taking $p$ bits. Using $|x|$ to mean the size of integer $x$, a float represents the value $m \cdot 2^{e-|m|+1}$.[4] Following Appendix A of Merrill & Sabharwal (2023), we define $p$-truncated addition $(+, \sum)$, multiplication $(\cdot)$, and division $(/)$ over $\mathbb{D}_p$. We now define a *transformer encoder binary classifier* over $\mathbb{D}_p$, largely adopting Merrill & Sabharwal's notation.[5]

**Definition 1.** A $p$-precision transformer $\mathcal{T}$ with $h$ heads, $d$ layers, model dimension $m$ (divisible by $h$), and feedforward width $w$ is specified by:

1. An embedding function $\phi : \Sigma \times \mathbb{N} \to \mathbb{D}_p^m$ whose form is defined in Appendix C.1;[6]
2. For each $1 \leq \ell \leq d$ and $1 \leq k \leq h$, a head similarity function $s_k^\ell : \mathbb{D}_p^m \times \mathbb{D}_p^m \to \mathbb{D}_p$ whose form is defined in Appendix C.2;
3. For each $1 \leq \ell \leq d$ and $1 \leq k \leq h$, a head value function $v_k^\ell : \mathbb{D}_p^m \to \mathbb{D}_p^{m/h}$ whose form is defined in Appendix C.2;
4. For each $1 \leq \ell \leq d$, an activation function $f^\ell : (\mathbb{D}_p^{m/h})^h \times \mathbb{D}_p^m \to \mathbb{D}_p^m$ whose form is defined in Appendix C.3 and implicitly uses the feedforward dimension $w$;
5. An output classifier head $\kappa : \mathbb{D}_p^m \to \{0, 1\}$ whose form is defined in Appendix C.4.

**Definition 2.** We define the transformer computation and output as a function of an input $x \in \Sigma^n$.

1. Embeddings: For $1 \leq i \leq n$, $\mathbf{h}_i^0 = \phi(x_i, i)$.[6]
2. Self Attention: For $0 \leq \ell \leq d - 1$, (multihead) self-attention block $\ell + 1$ computes $h$ attention heads:

$$\mathbf{a}_{i,k}^{\ell+1} = \sum_{j=1}^{n} \frac{s_k^{\ell+1}(\mathbf{h}_i^\ell, \mathbf{h}_j^\ell)}{Z_{i,k}} \cdot v_k^{\ell+1}(\mathbf{h}_j^\ell), \qquad \text{where } Z_{i,k} = \sum_{j=1}^{n} s_k^{\ell+1}(\mathbf{h}_i^\ell, \mathbf{h}_j^\ell).$$

3. Activation Block: For $0 \leq \ell \leq d - 1$, activation block $\ell + 1$ aggregates the head outputs to produce $\mathbf{h}^{\ell+1}$:

$$\mathbf{h}_i^{\ell+1} = f^{\ell+1}(\mathbf{a}_{i,1}^{\ell+1}, \ldots, \mathbf{a}_{i,h}^{\ell+1}, \mathbf{h}_i^\ell).$$

4. Classifier Head: The network prediction on $x \in \Sigma^n$ is $\kappa(\mathbf{h}_n^d)$.

---

[4] $\langle 101, 010 \rangle$ represents $1.01_2 \times 2^{10_2}$. This is closer to the IEEE standard than the $m \cdot 2^e$ semantics used in Merrill & Sabharwal (2023), letting us define the minimum representable float more realistically in Proposition 1.

[5] Increasing the classifier's output space arity (e.g., a transformer that predicts the next token) or switching to causal attention of a decoder-only model would not change our results. However, our proof no longer goes through if the decoder can generate tokens that get added to the input at the next step (cf. Pérez et al., 2019).

[6] $\phi$, like $p$, is actually a function of the context length $n$, and Appendix C.1 enforces that $\phi$ is computable in $\mathrm{O}(\log n)$ time, as standard choices of positional embeddings would satisfy.

We say $\mathcal{T}(x) = \kappa(\mathbf{h}^d_{|x|})$ and $L_\mathcal{T}$ is the language of $x \in \Sigma^*$ such that $\mathcal{T}(x) = 1$. We refer to $\phi, s^\ell_k, v^\ell_h, f^\ell$, and $\kappa$ as the **core functions** in $\mathcal{T}$, and to embeddings, self attention, activation, and the classifier head as the **components** of $\mathcal{T}$. We write $\theta_\mathcal{T}$ for the concatenated vector of parameters for the functions $\phi, s^\ell_k, v^\ell_h, f^\ell$, and $\kappa$, for all $1 \leq \ell \leq d$ and $1 \leq k \leq h$.

We define a **log-precision transformer** as one where $p$ is at most $\mathrm{O}(\log n)$ and is a "simple" function, i.e., computable in $\mathrm{O}(\log n)$ time. In our model, the weights $\theta_\mathcal{T}$ defining $\mathcal{T}$ are fixed, but the precision $p$ used to compute the forward pass can depend on $n$ (see Footnote 13 for a generalization).

## 2.2 First-Order Logic with Majority

As we will show, transformers can be translated into sentences in $\mathsf{FO}(\mathsf{M})$. But what do such sentences look like? Informally, $\mathsf{FO}(\mathsf{M})$ is first-order logic extended to also have majority ($\mathsf{M}$) quantifiers. Following Barrington et al. (1990), our sense of $\mathsf{FO}(\mathsf{M})$ takes *strings* in $\Sigma^*$ as input and returns 0 or 1 to define a formal language. In this setting, quantifiers range over *indices* (positions) into the string. Predicates can be applied to the variables introduced by these quantifiers.

**Definition 3** ($\mathsf{FO}(\mathsf{M})$ index). Indices in $\mathsf{FO}(\mathsf{M})$ are integers denoting positions in the input string:

1. The constant 1, representing the first token's position.
2. The constant $n$, representing the last token's position.
3. Strings (e.g., $i, j, k$) representing variables ranging over positions 1 to $n$.
4. Any index built by applying addition or subtraction to other indices.[7]

**Definition 4** ($\mathsf{FO}(\mathsf{M})$ formula). Formulas in $\mathsf{FO}(\mathsf{M})$ are constructed as follows:[8]

1. Let $\Sigma$ be a finite alphabet. For each $\sigma \in \Sigma$ and any index $i$, $\sigma(i)$, e.g., $\mathsf{a}(i)$, is a formula that is true if the $i$-th input token is $\sigma$.[9]
2. For any indices $i, j$, the formula $\mathrm{bit}(i, j)$ returns the $j$-th bit of the binary expansion of $i$.[10]
3. For two indices $i, j$, $i = j$, $i \leq j$, and $i \geq j$ are formulas with their conventional semantics.
4. For two formulas $\phi, \psi, \phi \wedge \psi$ and $\phi \vee \psi$ are formulas with their conventional semantics.
5. For any formula $\phi$ (which may refer to $i$), the following are valid formulas:
   (a) $\exists i.\ \phi$ means some value of $i$ in $[1, n]$ makes $\phi$ true.
   (b) $\forall i.\ \phi$ means all values of $i$ in $[1, n]$ make $\phi$ true.
   (c) $\mathsf{M}i.\ \phi$ means $\geq n/2$ values of $i$ in $[1, n]$ make $\phi$ true.

We use parentheses where necessary to disambiguate the order of operations. General formulas may contain free (i.e., unbound) variables: e.g., $\forall i.\ i = j$. A *sentence* is an $\mathsf{FO}(\mathsf{M})$ formula $\phi$ with no free variables. Sentences represent functions from from $\Sigma^*$ to $\{0, 1\}$ and thus define a formal language.[11]

**Extensions.** Beyond Definition 4, $\mathsf{FO}(\mathsf{M})$ can express *counting* and *threshold* quantifiers in terms of majority quantifiers (Barrington et al., 1990). Given a formula $\phi$, a counting quantifier creates a new formula $\exists^k i : \phi$ that is true iff $\phi$ is true across exactly $k$ values of $i$. Threshold quantifiers $\exists^{\leq k}$ and $\exists^{\geq k}$ work similarly but check if $\phi$ is true for at least or at most $k$ values of $i$. In addition, we show in Appendix A that $\mathsf{FO}(\mathsf{M})$ can express *conditional majority* quantifiers, which create a formula $\mathsf{M}i : \phi\ [\psi]$ that is true iff $\psi$ is true for at least half the values of $i$ that make $\phi$ true.

### 2.2.1 Examples

To illustrate the formalism, we provide example languages definable in $\mathsf{FO}(\mathsf{M})$ with $\Sigma = \{\mathsf{a}, \mathsf{b}\}$. First, we show two languages that do not require majority quantifiers to express:

**Example 1** (Bigram matching). Strings containing the bigram $\mathsf{ab}$: $\exists i\ [\mathsf{a}(i) \wedge \mathsf{b}(i + 1)]$.

**Example 2** (Skip-bigram matching). Strings containing the long-distance pattern $\mathsf{a} \ldots \mathsf{b}$ (cf. "induction heads" of Elhage et al. 2021): $\exists i\ [\mathsf{b}(i) \wedge \exists j\ [j \leq i \wedge \mathsf{a}(j)]]$.

---

[7]Barrington et al. (1990) did not introduce this as a primitive, but it can be simulated using the $\leq$ predicate.

[8]We write parentheses to indicate the order of operations.

[9]Barrington et al. (1990) define $Q_b(i)$ for $b \in \{\mathsf{0}, \mathsf{1}\}$. We generalize this to an arbitrary vocabulary $\Sigma$ by assuming each token is one-hot-encoded: $\sigma(i) = Q_1(|\Sigma| i + s)$ where $s$ is the index of $\sigma$ in the vocabulary.

[10]This predicate is included in the logic for technical reasons; see Barrington et al. (1990).

[11]One can also take multiple sub-sentences within $\phi$ to be labeled as ordered outputs, thus allowing $\phi$ to be a function from $\Sigma^*$ to $\{0, 1\}^k$ for some fixed constant $k$.

In contrast, Example 3 is a simple example that requires majority quantifiers (Furst et al., 1984):

**Example 3** (Majority). Strings with more b's than a's: $\mathsf{M}i\,[\mathtt{b}(i)]$.

Figure 1 showed how $\mathsf{FO}(\mathsf{M})$ can be used to recognize patterns like $\mathtt{a}^m\mathtt{b}^m$. A similar idea can be used to model parentheses matching (Barrington et al., 1990):

**Example 4** (1-Dyck). The well-balanced parentheses language (with $\mathtt{a}$ opening and $\mathtt{b}$ closing):

$$\forall i.\,(\exists a, b.\,((\exists^a j : \mathtt{a}(j) \wedge j \leq i) \wedge (\exists^b j : \mathtt{b}(j) \wedge j \leq i) \wedge b \leq a)) \wedge \mathsf{M}i.\,\mathtt{a}(i) \wedge \mathsf{M}j.\,\mathtt{b}(j).$$

**Example 5** (Integer Arithmetic). Iterated addition (i.e., summing $n$ $n$-bit numbers), iterated multiplication, and division (Hesse, 2001) can all be expressed in $\mathsf{FO}(\mathsf{M})$.

## 3  Finite Precision Transformers Cannot Attend Universally

Attention heads that spread attention weight uniformly across inputs have been observed in transformer LMs (Merrill et al., 2021) and make soft attention fundamentally more powerful than hard attention (Hao et al., 2022; Merrill et al., 2022). In particular, uniform attention is an important primitive that transformers can use to solve tasks involving counting (Bhattamishra et al., 2020; Chiang et al., 2023), taking majority votes (Merrill et al., 2022), and matching parentheses or sorting (Weiss et al., 2021). A transformer with sufficient precision can easily implement uniform attention by setting the keys and queries across all positions to be constant. However, attention heads with finite precision cannot represent uniform attention over long sequences as a consequence of the following:

**Proposition 1.** *Let $\mathbf{a} \in \mathbb{R}^n$ s.t. $\sum_{i=1}^n a_i = 1$ and $\tilde{\mathbf{a}}$ its nearest $p$-precision float approximation.*

1. *Then the number of nonzero entries of $\tilde{\mathbf{a}}$ is upper bounded by its precision: specifically, $\tilde{\mathbf{a}}$ has at most $2^{2^p}$ nonzero entries.*
2. *Moreover, if $p < \log\log n$ and $\mathbf{a}$ is uniform (i.e., $a_i = 1/n$), then $\tilde{\mathbf{a}} = \vec{0}$.*

*Proof.* The smallest positive value representable by a $p$-precision float is $2^{-(p_m - 2 + 2^{p_e - 1})}$ which is bounded below by $2^{-2^p + 1}$. Letting $k = 2^{2^p}$, it holds that $2^{-2^p + 1} = 2/k$. So if $\tilde{a}_i$ gets the minimum value, then $a_i \geq 1/k$. Since $\sum_i a_i = 1$, there can be at most $k$ indices satisfying this property. This implies there can be at most $k$ nonzero entries in $\tilde{\mathbf{a}}$. If $n > k$ and $\mathbf{a}$ is uniform, $1/n$ is less than half of the minimum representable value of $2/k$. Thus, $\tilde{\mathbf{a}} = \vec{0}$. □

Proposition 1 says that fixed-precision transformers are artificially limited because they can only attend over bounded-length windows, making them similar to hard-attention transformers (Hao et al., 2022). Morever, they cannot compute uniform attention over contexts of length $n$ with less than $\log\log n$ precision. This explains why Chiang et al. (2023) prove finite-precision transformers provably cannot recognize $\mathtt{a}^m\mathtt{b}^m$, while in practice transformers have been shown to learn even its harder variant $\mathtt{a}^m\mathtt{b}^m\mathtt{c}^m$ even with long context lengths (Bhattamishra et al., 2020). In essence, their upper bound only applies in the asymptotic regime when $n > 2^{2^p}$.

In contrast, transformers in practice have enough precision both to compute uniform attention and recognize $\mathtt{a}^m\mathtt{b}^m$ on practical context lengths. More concretely, the bfloat16 representation allows uniform attention over $2^{6+2^7} \approx 10^{42}$ tokens and normal float16[12] allows $2^{10+2^4} \approx 10^8$ tokens, both well above the typical context window of transformers. This motivates a formal model of transformers with enough precision to compute uniform attention and recognize languages such as $\mathtt{a}^m\mathtt{b}^m$.

## 4  Main Result: Expressing Log-Precision Transformers in $\mathsf{FO}(\mathsf{M})$

By Proposition 1, precision must grow with the context length $n$ ($p > \log\log n$) for a transformer to compute uniform attention and other attention patterns with unbounded range, like practical transformers. In this paper, we analyze any transformer with up to $O(\log n)$ precision. We show that any function computable by log-precision transformers can be expressed in $\mathsf{FO}(\mathsf{M})$:

---

[12]We account for the division of $p$ into $p_m$ and $p_e$ rather than treating them together. Our minimum value differs slightly from numpy but is on the same order of magnitude. Moving to float8 lowers the length upper bound for uniform attention to $2^{3+2^3} \approx 2048$, which suggests float8 LMs will have limited length generalization.

**Theorem 2.** *Let $\mathcal{T}$ be a log-precision transformer with a parameter vector $\theta_{\mathcal{T}}$ fixed for all context lengths $n$.*[13] *Then, there exists an* FO(M) *sentence $\phi$ that computes the same function as $\mathcal{T}$, i.e., $\phi(x) = \mathcal{T}(x)$ for any input string $x$.*

Theorem 2 is the tightest known upper bound for log-precision transformers and shows that it is still possible to characterize transformers in a simple variant of first-order logic even with log-precision and uniform attention. As alluded to earlier, Theorem 2 immediately implies that any problem complete for FO(M) (or a larger class) is also transformer-hard. Since integer division and Dyck language membership are known to be FO(M)-complete (Hesse, 2001; Aaronson et al., 2022), it follows, perhaps surprisingly, that the entire computation of any transformer on input $x$ can be reduced to a single integer division or a finite number of Dyck-language queries:

**Corollary 2.1.** *Let $\mathcal{T}$ be a transformer satisfying Theorem 2. For any input $x$, there exist first-order definable integers $a, b$, and $i$ (dependent on $\mathcal{T}$ and $x$) such that $\mathcal{T}(x)$ equals the $i$-th bit of $\lfloor a/b \rfloor$. For any $x$, there also exist first-order definable strings $w_1, \ldots, w_m$ such that $\mathcal{T}(x)$ is first-order definable in terms of the membership of the $w_i$'s in $k$-Dyck.*

# 5 Preliminaries for Proving Theorem 2

## 5.1 Computation Graphs

A *computation graph* $G$ over a datatype $\mathbb{D} \subseteq \{0, 1\}^*$ and a countable set of primitive functions $\mathfrak{F} \subseteq \mathbb{D}^* \times \mathbb{D}$ is a directed acyclic graph where:

1. Each node is labelled by a *node type*: a function $f \in \mathfrak{F}$ computed by this node.
2. Each edge represents a value $\mathbb{D}$ flowing as output from one node into another node. We consider the edges flowing into node $j$ to have an order, i.e., be numbered.
3. $\mathfrak{F}$ contains the special symbol input, which designates $k$ nodes as input nodes. We refer to $k$ as the *arity* and assume w.l.o.g. that nodes $0, \ldots, k-1$ are inputs.[14]
4. A single node is taken as the output node (w.l.o.g., the node with the largest index).

A computation graph $G$ of arity $k$ parameterizes a function $\mathbb{D}^k \to \mathbb{D}$ in the standard way: the input nodes are assigned the input values, and the value of each node is computed (traversing the graph in a bottom-up topological order) as a function of the values of its children until the output node receives a value. The value of the output node is considered the output of the function. It is worth noting that computation graphs can only process inputs of bounded length. To process arbitrary-length inputs, we will need to generalize them to computation graph families (Section 5.2).

For a computation graph $G$, size($G$) is the number of nodes, depth($G$) is the length of the longest path from an input node to the output, and arity($G, i$) is the number of inputs to node $i$.

**Threshold circuits.** A threshold circuit is a special case of a computation graph where $\mathbb{D} = \{0, 1\}$ and $\mathcal{F}$ is the set of threshold functions of the form $\theta_{\leq \Delta}$ and $\theta_{\geq \Delta}$ over $\mathbb{D}^*$, defined as follows: $\theta_{\leq \Delta}(x) = 1$ if $\sum_{\sigma \in x} \sigma \leq \Delta$ and 0 otherwise; $\theta_{\geq \Delta}(x)$ is defined analogously. Typical AND, OR, and NOT gates are a special case of threshold gates, as is an IDENTITY gate.[15]

We allow nodes with the $k' \geq 1$ largest indices to all be designated as (ordered) output nodes. A threshold circuit with arity $k$ and $k'$ output nodes will thus be a function from $\{0, 1\}^k$ to $\{0, 1\}^{k'}$. This will be convenient when simulating neural network components that output multiple bits.

We will find it useful to consider threshold circuits as a kind of compilation target for computation graphs: in other words, we will be concerned with simulating computation graphs defined over more complex functions and data types into threshold circuits.

## 5.2 Computation Graph Families

A computation graph family over $\mathbb{D}$ and $\mathfrak{F}$ is a mapping from $n \in \mathbb{N}$ to a computation graph $G_n$ for processing inputs of size $n$. Thus, $\mathcal{G}$ defines a function from $\mathbb{D}^* \to \mathbb{D}$, where $\mathcal{G}(x) = G_{|x|}(x)$.

---

[13]Theorem 2 can also be extended to apply to log-precision transformers with *log-uniform weights*, i.e., where $\theta_{\mathcal{T}}$ can grow in size and precision with $n$ (see Appendix B).

[14]By convention in computer science, we let computation graph nodes be zero-indexed.

[15]For more background on threshold circuits, see Merrill & Sabharwal (2023) and Merrill et al. (2022).

Intuitively, computation graph families are useful because they generalize computation graphs to define functions over *unbounded-length* strings as inputs.

**Size, depth, and arity.** For computation graph families, the size, depth, and arity become functions of the input length $n$: $\mathsf{size}_\mathcal{G}(n) = \mathsf{size}(G_n), \mathsf{depth}_\mathcal{G}(n) = \mathsf{depth}(G_n), \mathsf{arity}_\mathcal{G}(n, i) = \mathsf{arity}(G_n, i)$.

**Uniformity.** The infinite set $\mathcal{G}$ can be alternatively represented by two functions:

1. $\mathsf{node}_\mathcal{G}(n, i)$, which returns the type of node $i$ in $G_n$ if $i \leq \mathsf{size}(G_n)$, and $\emptyset$ otherwise. For example, if node $i$ computes the logical AND of its inputs, then $\mathsf{node}_\mathcal{G}(n, i) = \wedge$.
2. $\mathsf{edge}_\mathcal{G}(n, i, j)$, which returns the argument index of $i$ into node $j$ if $G_n$ contains an edge $i \rightarrow j$ and $-1$ otherwise. $\mathsf{edge}_\mathcal{G}(n, i, j)$ only needs to be defined over $i, j < \mathsf{size}(G_n)$. For example, if $G_n$ contains a node $j$ with three incoming edges, the second of which comes from node $i$, then $\mathsf{edge}_\mathcal{G}(n, i, j) = 1$.

A pair of algorithms implementing these two functions uniquely specifies a computation graph family, as it enables building the computation graph $G_n$ for any $n$. Uniform computation graph families (generalizing uniform circuits; cf. Arora & Barak, 2009) are families where $\mathsf{node}_\mathcal{G}$ and $\mathsf{edge}_\mathcal{G}$ can be computed efficiently, i.e., under some constraints on space or time:

**Definition 5** (Uniformity). A computation graph family $\mathcal{G}$ is $T(n)$-uniform iff $\mathsf{node}_\mathcal{G}(n, i)$ and $\mathsf{edge}_\mathcal{G}(n, i, j)$ can be computed by a deterministic Turing machine in time $T(n)$. We focus on *log-uniform* computation graph families: i.e., where $T(n) = \mathrm{O}(\log n)$.[16]

**Threshold circuit families.** These are simply families of threshold circuits. We will be simulating computation graph families with threshold circuit families. Log-uniform $\mathsf{TC}^0$ is the class of languages recognized by log-uniform constant-depth, poly-size threshold circuit families. See Merrill & Sabharwal (2023); Liu et al. (2023); Arora & Barak (2009) for more background on $\mathsf{TC}^0$ and circuits.

## 6 Proof of Theorem 2

The idea is to simulate a transformer with a log-uniform $\mathsf{TC}^0$ circuit family. Since log-uniform $\mathsf{TC}^0 = \mathsf{FO}(\mathsf{M})$, this would imply any transformer can be expressed in $\mathsf{FO}(\mathsf{M})$. First, we note that transformers are log-uniform computation graphs:

**Lemma 1** (Proof in Appendix B.1). *A transformer $\mathcal{T}$ is a log-uniform computation graph family where $\mathfrak{F}$ contains embedding, self-attention, feedforward, and output components.*

Further, each core module of the transformer can be simulated by a log-uniform $\mathsf{TC}^0$ circuit family:

**Lemma 2** (Proof in Appendix B.2). *Let $\mathcal{T}$ be a log-precision transformer with fixed parameters $\theta_\mathcal{T}$. Then each component in $\mathfrak{F}$ is computable in log-uniform $\mathsf{TC}^0$.*

Intuitively, we can now simulate a transformer in log-uniform $\mathsf{TC}^0$ by just simulating each of its components with a threshold circuit and routing their inputs and outputs appropriately. However, we will need two more technical conditions to verify that this construction is indeed log-uniform:

**Lemma 3** (Proof in Appendix B.3). *Let $\mathcal{T}$ be a log-precision transformer with fixed parameters $\theta_\mathcal{T}$. There exists a function $\mathsf{bsize}(n)$ that is a power of $2$ and computable in $\mathrm{O}(\log n)$ time s.t. $\mathsf{size}_\mathcal{F}(n) \leq \mathsf{bsize}(n)$ for all $\mathcal{F} \in \mathfrak{F}$.*

**Lemma 4** (Proof in Appendix B.4). *If $\mathcal{F}$ is a log-uniform $\mathsf{TC}^0$ family and $\mathsf{size}_\mathcal{F}(n) \leq \mathsf{bsize}(n)$, there exists a log-uniform $\mathsf{TC}^0$ family $\mathcal{F}'$ s.t. $\mathcal{F}(x) = \mathcal{F}'(x)$ for all $x$ and $\mathsf{size}_{\mathcal{F}'}(n) = \mathsf{bsize}(n)$.*

Combined, Lemmas 3 and 4 show that each $\mathcal{F} \in \mathfrak{F}$ is computable by a log-uniform $\mathsf{TC}^0$ family with size $\mathsf{bsize}(n)$ that is a power of 2 and computable in time $\mathrm{O}(\log n)$. We will show these conditions imply a transformer $\mathcal{T}$ can be simulated by a $\mathsf{TC}^0$ family $\mathcal{C}$ (Theorem 3) and moreover that $\mathcal{C}$ is log-uniform (Corollary 3.2). By the equivalence of log-uniform $\mathsf{TC}^0$ and $\mathsf{FO}(\mathsf{M})$ (Barrington et al., 1990), we then conclude that any log-precision transformer can be expressed in $\mathsf{FO}(\mathsf{M})$.

---

[16]Past work (Merrill & Sabharwal, 2023) analyzes transformers with a similarly named but weaker notion of uniformity, namely log-*space* (rather than log-*time*) uniformity.

| **Algorithm 1** $\mathsf{node}_{\mathcal{C}}(n, i)$ | **Algorithm 2** $\mathsf{edge}_{\mathcal{C}}(n, i, j)$ |
|---|---|
| *Return the type of gate $i$ in circuit $C_n$.* | *If $C_n$ contains an edge $i \to j$, return the argument number of that edge. Otherwise, return $-1$.* |

| | |
|---|---|
| 1: $\mathcal{F} \leftarrow \mathsf{node}_{\mathcal{G}}(n, \mathsf{bnode}(n, i))$ | 1: $i' \leftarrow \mathsf{bnode}(n, i)$ |
| 2: **if** $\mathcal{F} \neq \emptyset$ **then** | 2: $j' \leftarrow \mathsf{bnode}(n, j)$ |
| 3: $\quad$ **return** $\mathsf{node}_{\mathcal{F}}(n, i - \mathsf{bstart}(n, i'))$ | 3: $s_i \leftarrow \mathsf{bstart}(n, i')$ |
| 4: **else return** $\emptyset$ | 4: $s_j \leftarrow \mathsf{bstart}(n, j')$ |
| | 5: **if** $i' = j'$ **then** |
| | 6: $\quad \mathcal{F} \leftarrow \mathsf{node}_{\mathcal{G}}(n, i')$ |
| | 7: $\quad$ **return** $\mathsf{edge}_{\mathcal{F}}(n, i - s_i, j - s_j)$ |
| | 8: **else if** $\mathsf{edge}_{\mathcal{G}}(n, i', j') \geq 0$ **then** |
| | 9: $\quad b_i \leftarrow i - (s_i + \mathsf{bsize}(n, i') - p(n))$ |
| | 10: $\quad b_j \leftarrow j - (s_j + p(n) \cdot \mathsf{edge}_{\mathcal{G}}(n, i', j'))$ |
| | 11: $\quad$ **if** $b_i = b_j < p(n)$ **then return** $j - s_j$ |
| | 12: $\quad$ **else return** $-1$ |
| | 13: **else return** $-1$ |

## 6.1 Simulating Computation Graph Families with Circuit Families

We give algorithms that take a computation graph family and define a circuit family simulating it. Intuitively, the algorithms creates contiguous blocks of circuit gates simulating each node in the computation graph and route inputs and outputs between blocks appropriately.

**Block mapping.** This algorithm depends on a *block mapping*, which is an implementation of the following three functions:

1. The *block node* $\mathsf{bnode}(n, i)$: the index of the node that gate $i$'s block is simulating.
2. The *block start* $\mathsf{bstart}(n, i')$: the smallest gate index in the block simulating node $i'$.
3. The *block size* $\mathsf{bsize}(n, i')$: the number of gates in the block simulating node $i'$.

Further, we enforce that a valid block mapping must satisfy that, for all $i$, with $i' = \mathsf{bnode}(n, i)$,

$$\mathsf{bstart}(n, i') \leq i < \mathsf{bstart}(n, i') + \mathsf{bsize}(n, i').$$

Let $\mathcal{G}$ be a computation graph whose primitive functions are computable by log-uniform threshold circuits. We can identify each primitive function with a log-uniform threshold circuit family $\mathcal{F}$ that computes it, where the first $\mathsf{arity}_{\mathcal{F}}(n)$ gates are IDENTITY gates reserved for taking input. For such a graph, $\mathsf{node}_{\mathcal{G}}$ can be taken to return a symbol identifying a circuit family $\mathcal{F}$. In this case, our algorithm requires that, for all $i'$, the block size of $i'$ must match the size of the circuit for the type of block $i'$, i.e., $\mathsf{bsize}(n, i') = \mathsf{size}_{\mathsf{node}_{\mathcal{G}}(n, i')}(n)$. These properties let us meaningfully identify a graph node $i'$ with a block of nodes that will simulate it. This intuition enables us to develop Algorithms 1 and 2 for constructing a uniform threshold circuit family from a uniform computation graph family.

**Theorem 3.** *Let $\mathcal{G}$ be a computation graph over a finite set of node types $\mathfrak{F}$, where each $\mathcal{F} \in \mathfrak{F}$ is specified by a log-uniform circuit family. Let $\mathsf{bnode}$, $\mathsf{bstart}$, and $\mathsf{bsize}$ be a valid block mapping in the sense above. Then Algorithms 1 and 2 define a circuit family $\mathcal{C}$ such that*

1. *$\mathcal{C}$ and $\mathcal{G}$ compute the same $\mathbb{D}_p^* \to \mathbb{D}_p$ function (let the final $p$ gates of each $C_i$ be its output).*
2. *$\mathsf{depth}_{\mathcal{C}}(n) \leq \mathsf{depth}_{\mathcal{G}}(n) \cdot \max_{\mathcal{F}} \mathsf{depth}_{\mathcal{F}}(n)$.*
3. *$\mathsf{size}_{\mathcal{C}}(n) \leq \mathsf{size}_{\mathcal{G}}(n) \cdot \max_{\mathcal{F}} \mathsf{size}_{\mathcal{F}}(n)$.*

*Proof.* Assume w.l.o.g. that the gates of $\mathcal{C}$ are topologically ordered. We show by induction over circuit gates $j$ (with $j' = \mathsf{bnode}(n, j)$) that:

1. For all $i' < j'$, the last $p$ nodes of block $i'$ store the value of node $i'$.
2. For all $i$ such that $\mathsf{bstart}(n, j') \leq i \leq j$, gate $i$ of $\mathcal{C}$ (as a function of the input nodes of $j'$) computes gate $i - \mathsf{bstart}(n, j')$ of $\mathsf{node}_{\mathcal{G}}(n, j')$.

Base case. We have two circuits with no gates, so the premises are trivially satisfied.

Inductive case. Assume the premises hold up to $j$. We will show they hold for $j + 1$. Let $\mathcal{T} = \mathsf{node}_{\mathcal{G}}(n, j')$. By Premise 1, we know that the last $p$ nodes of block $i'$ store the output of node $i'$, for

$i' < j'$. By Algorithm 2, for each $i'$ such that $\mathsf{edge}_\mathcal{G}(n, i', j') = a$ with $0 \leq k < \mathsf{arity}_\mathcal{F}(n)$, gates $kp$ through $k(p+1) - 1$ of block $j'$ will copy the final $p$ gates of block $i'$. Thus, the first $k \cdot \mathsf{arity}_\mathcal{F}(n)$ gates of block $j'$ store the inputs to node $j'$.

At this point, we use Premise 2 to conclude that the first $j - \mathsf{bstart}(n, j')$ gates of block $j'$ compute the same function as the first $j - \mathsf{bstart}(n, j')$ gates of $\mathcal{F}$ with respect to this input. Thus, we just need to show that gate $j + 1$ is also correct. Within Algorithm 2, we fall in case $i' = j'$, meaning that gate $j + 1$ of block $j'$ gates the same inputs as gate $j + 1$ of $\mathcal{F}$. By Algorithm 1, the type of gate $j + 1$ in block $j'$ is the type of gate $j + 1$ of $\mathcal{F}$. Thus, gate $j + 1$ in block $j'$ computes the same function of the input gates as gate $j + 1$ in $\mathcal{F}$. If $j + 1 = \mathsf{bsize}(n, j')$, we conclude that the final $p$ gates of block $j'$ store the output of node $j'$. $\qquad\square$

Let $\mathsf{XC}^0$ denote any family of constant-depth, poly-size circuits, including $\mathsf{AC}^0$ and $\mathsf{TC}^0$.[17]

**Corollary 3.1.** *Let $\mathcal{G}$ be a constant-depth, poly-size computation graph family over a finite $\mathfrak{F}$. If every node type in $\mathfrak{F}$ can be computed by $\mathsf{XC}^0$ circuits, the function computed by $\mathcal{G}$ is in $\mathsf{XC}^0$.*

Since a transformer has constant depth and polynomial size, Corollary 3.1 lets us easily recover prior results about hard-attention transformers (Hao et al., 2022; Hahn, 2020) and saturated attention transformers (Merrill et al., 2022) using a common framework. All one has to do is show that all individual node types in such transformers can be computed by $\mathsf{AC}^0$ and $\mathsf{TC}^0$ circuits, respectively.

Corollary 3.1 established that Algorithms 1 and 2 construct a circuit family that simulates $\mathcal{G}$. With the right block mapping, $\mathcal{C}$ will be log-uniform as long as $\mathcal{G}$ and its node types are log-uniform.

**Corollary 3.2.** *Let $\mathcal{G}$ be a log-uniform, constant-depth computation graph family over a finite $\mathfrak{F}$, where each $\mathcal{F} \in \mathfrak{F}$ is specified by a log-uniform $\mathsf{TC}^0$ family with $\mathsf{size}_\mathcal{F}(n) = \mathsf{bsize}(n)$ that is a power of 2 computable in $\mathrm{O}(\log n)$ time. Then $\mathcal{G}$ can be simulated by a log-uniform $\mathsf{TC}^0$ family $\mathcal{C}$ that obeys the size and depth properties of Theorem 3.*

*Proof.* Let $\mathcal{C}$ be the circuit family defined by Algorithms 1 and 2 given $\mathcal{G}$ and the following block mapping: $\mathsf{bnode}(n, i) = \lfloor i/\mathsf{bsize}(n) \rfloor, \mathsf{bstart}(n, i') = i' \cdot \mathsf{bsize}(n), \mathsf{bsize}(n, i') = \mathsf{bsize}(n)$. Since $\mathsf{bsize}(n)$ is a power of 2, bnode and bstart are reducible to left and right shifting over $\mathrm{O}(\log n)$-bit integers, which can be implemented in $\mathrm{O}(\log n)$ time. Thus, each block mapping function is computable in time $\mathrm{O}(\log n)$. Since $\mathsf{node}_\mathcal{G}$ and $\mathsf{edge}_\mathcal{G}$ are just calling functions computable in time $\mathrm{O}(\log n)$ with constant overhead, we conclude that $\mathcal{C}$, the circuit family they define, is log-uniform, and it is already known to simulate $\mathcal{G}$ with constant depth and polynomial size by Theorem 3. $\quad\square$

# 7  Conclusion

We proved that any log-precision transformer classifier can be translated to an $\mathsf{FO}(\mathsf{M})$ sentence that computes the same function (on all inputs of any length). This result comes by first simulating a transformer with a highly uniform threshold circuit family, and then leveraging the established equivalence of log-uniform circuits and $\mathsf{FO}(\mathsf{M})$. Transformers and other neural nets are often discussed in contrast with symbolic models based on logical formalisms (Garnelo & Shanahan, 2019)—an immediate implication of our result is that it is possible to express the inner workings of transformers also in a simple logic, challenging the premise of a rigid division between symbolic and neural models. Our results also provide the tightest known upper bound on log-precision transformers.

While it is striking that a full transformer can be translated to a sentence in a logic as simple as $\mathsf{FO}(\mathsf{M})$, we believe the bound is not tight. In particular, we conjecture that it is possible to simulate any transformer with an $\mathsf{FO}(\mathsf{M})$ sentence of quantifier depth of at most 2, which could be proven by establishing a hierarchy theorem describing the $\mathsf{FO}(\mathsf{M})$ quantifier depth needed to simulate a $\mathsf{TC}^0$ family of a certain size. It would also be an interesting extension to translate real transformers to $\mathsf{FO}(\mathsf{M})$ sentences. In this sense, we believe our results provide a theoretical foundation to guide mechanistic interpretability work (cf. Weiss et al., 2021; Lindner et al., 2023).

Our findings provide a novel view into transformer classifiers and their limits. It would be exciting for future research to extend our results to account for other common practical uses of transformers, such as for long-form generation, chain-of-thought reasoning, and in-context learning.

---

[17]Formally, $\mathfrak{F}$ just needs to contain $\wedge$ and $\vee$.

**Acknowledgments**

We thank Paul Beame, David Chiang, anonymous reviewers, and researchers at the Allen Institute for AI for feedback. WM was supported by an NSF graduate research fellowship and in part by NSF award 1922658.

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
