## A  Conditional Majority

Given formulas $\phi, \psi$, $Mi : \phi.$ $\psi$ is a sentence that is true iff $\psi$ is true for at least half the values of $i$ that make $\phi$ true.

**Proposition 2.** *For any two predicates $\phi(i)$ and $\psi(i)$, $Mi : \phi(i).$ $\psi(i)$ can be expressed in $\mathsf{FO}(\mathsf{M})$.*

*Proof.* $Mi : \phi.$ $\psi$ can be rewritten using a counting quantifier and a threshold quantifier:

$$\exists k, k'. \left[ 2k' = k \wedge \exists^k i : \phi(i) \wedge \exists^{\geq k'} j : (\phi(j) \wedge \psi(j)) \right].$$

The formula $2k' = k$ can be defined using bit. We then use the fact that counting and threshold quantifiers can be expressed in terms of majority quantifiers (Barrington et al., 1990) to conclude that $Mi : \phi.$ $\psi$ can be expressed in $\mathsf{FO}(\mathsf{M})$. $\qquad\square$

## B  Omitted Proofs

Table 1 summarizes the notation we use in the following proofs when describing computation graphs and circuit families.

Table 1:  Summary of common notation for computation graph and circuit families.

| Graph | Circuit | Output Range | Description |
|---|---|---|---|
| $i'$ | $i$ | $\mathbb{Z}$ | index of node or gate |
| $\mathsf{node}_{\mathcal{G}}(n, i')$ | $\mathsf{node}_{\mathcal{C}}(n, i)$ | $\mathfrak{F}^{18}$ | type of node or gate |
| $\mathsf{edge}_{\mathcal{G}}(n, i', j')$ | $\mathsf{edge}_{\mathcal{C}}(n, i, j)$ | $\mathbb{Z}$ | argument # of edge $i \rightarrow j$ |
| $\mathsf{size}_{\mathcal{G}}(n)$ | $\mathsf{size}_{\mathcal{C}}(n)$ | $\mathbb{Z}$ | # of nodes or gates |
| $\mathsf{depth}_{\mathcal{G}}(n)$ | $\mathsf{depth}_{\mathcal{C}}(n)$ | $\mathbb{Z}$ | longest path length |

| | | | |
|---|---|---|---|
| | $\mathsf{bnode}(n, i)$ | $[0, \mathsf{size}_{\mathcal{G}}(n)]$ | block containing $i$ |
| | $\mathsf{bstart}(n, i')$ | $[0, \mathsf{size}_{\mathcal{C}}(n)]$ | first gate in block $i'$ |
| | $\mathsf{bsize}(n, i')$ | $\mathbb{Z}$ | size of block $i'$ |

### B.1  Transformers are Log-Uniform Computation Graph Families

We now justify that the computation graph family defining a transformer is log-uniform. To do this, we introduce a stronger notion of uniformity called *column uniformity* that captures the highly regular structure of the transformer.

Let $\mathsf{node}(G, i)$ be the $i$-th node of computation graph $G$. Let $a \bmod b$ be the remainder when $a$ is divided by $b$.

**Definition 6** (Column uniformity). A computation graph family $\mathcal{G}$ is $T(n)$-column-uniform iff there exists a computation graph $K$ (with fixed size w.r.t $n$) such that, for all $i, j$ such that $0 \leq i, j < \mathsf{size}_{\mathcal{G}}(n)$:

1. $\mathsf{node}_{\mathcal{G}}(n, i) = \mathsf{node}\,(K, i \bmod \mathsf{size}(K))$.
2. If $\lfloor i/\mathsf{size}(K) \rfloor = \lfloor j/\mathsf{size}(K) \rfloor$, then

$$\mathsf{edge}_{\mathcal{G}}(n, i, j) = \mathsf{edge}\,(K, i \bmod \mathsf{size}(K), j \bmod \mathsf{size}(K)).$$

Otherwise, $\mathsf{edge}_{\mathcal{G}}(n, i, j)$ can be computed by a deterministic Turing machine in time $T(n)$.

We define *log-column-uniform* analogously to log-uniform: i.e., we let $T(n) = \mathrm{O}(\log n)$. log-column-uniform implies log-uniform because our implementations of $\mathsf{node}_{\mathcal{G}}$ and $\mathsf{edge}_{\mathcal{G}}$ can store $K$ in a finite lookup table and compute the quotient and remainder of $i$ and $j$ by $\mathsf{size}(K)$ in $\mathrm{O}(\log n)$ time using Lemma 12. The edges outside of $K$ are computable in $\mathrm{O}(\log n)$ time by construction.

**Lemma 1** (Proof in Appendix B.1). *A transformer $\mathcal{T}$ is a log-uniform computation graph family where $\mathfrak{F}$ contains embedding, self-attention, feedforward, and output components.*

*Proof.* We show the stronger condition that any transformer $\mathcal{T}$ is a log-column-uniform computation graph family, which implies it is log-uniform.

We have the column $K$ by Definition 2: all that remains to show is that $\mathsf{edge}_{\mathcal{G}_\mathcal{T}}$ can be computed in time $O(\log n)$ for edges outside the column. These edges route from the layer $\ell$ output to the self-attention heads of layer $\ell + 1$. Following from the column structure, there exists $k_\ell$ such that a node $i$ is an output vector of layer $\ell$ iff $k_\ell = i \bmod \mathsf{size}(K)$. In a finite lookup table, we can store $k_\ell$ for each $\ell + 1$, and use this for self-attention routing. For an unmasked self-attention head $j$, we compute:

$$\mathsf{edge}_{\mathcal{G}_\mathcal{T}}(n, i, j) = \begin{cases} \lfloor i/\mathsf{size}(K) \rfloor & \text{if } k_\ell = i \bmod \mathsf{size}(K) \\ -1 & \text{otherwise.} \end{cases}$$

For causally masked attention, we extend the first case to check that $\lfloor i/\mathsf{size}(K) \rfloor \leq \lfloor j/\mathsf{size}(K) \rfloor$. Either way, this logic can be implemented in time $O(\log n)$ via Lemma 12. Thus, we conclude that $\mathcal{G}_T$ is column-uniform. $\qquad\square$

## B.2 Transformer Components are Computable by Log-Uniform Threshold Circuits

**Lemma 2** (Proof in Appendix B.2). *Let $\mathcal{T}$ be a log-precision transformer with fixed parameters $\theta_\mathcal{T}$. Then each component in $\mathfrak{F}$ is computable in log-uniform $\mathsf{TC}^0$.*

We prove a more general version of Lemma 2 that handles some cases with weights growing with $n$. The weights $\theta_\mathcal{T}$ are just a special case of a computation graph (that do not depend on the input); we can thus apply our definition of log-uniform to them. Lemma 2 follows from a more general result with log-uniform $\theta_\mathcal{T}$:

**Lemma 5.** *Let $\mathcal{T}$ be a log-uniform transformer with log-uniform $\theta_\mathcal{T}$. Then each component in $\mathfrak{F}$ is computable in log-uniform $\mathsf{TC}^0$.*

*Proof.* In Appendix C, we show that log-uniform $\theta_\mathcal{T}$ implies:

1. The embedding component is computable in log-uniform $\mathsf{TC}^0$ (Lemma 6).
2. The self attention mechanism is computable in log-uniform $\mathsf{TC}^0$ (Lemma 7).
3. The activation block is computable in log-uniform $\mathsf{TC}^0$ (Lemma 8).
4. The output classifier head is computable in log-uniform $\mathsf{TC}^0$ (Lemma 9).

We have shown that each $\mathcal{F} \in \mathfrak{F}$ is computable in log-uniform $\mathsf{TC}^0$. $\qquad\square$

## B.3 Transformer Component Size Has a Log-Time Upper Bound

**Lemma 3** (Proof in Appendix B.3). *Let $\mathcal{T}$ be a log-precision transformer with fixed parameters $\theta_\mathcal{T}$. There exists a function $\mathsf{bsize}(n)$ that is a power of 2 and computable in $O(\log n)$ time s.t. $\mathsf{size}_\mathcal{F}(n) \leq \mathsf{bsize}(n)$ for all $\mathcal{F} \in \mathfrak{F}$.*

*Proof.* Let $2^{b(n)}$ be the least power of 2 at least as large as $\mathsf{size}_\mathcal{F}(n)$ for all $\mathcal{F}$. We observe that $2^{b(n)}$ is at most $2 \cdot \max_\mathcal{F} \mathsf{size}_\mathcal{F}(n)$ for all $n$. Because each $\mathcal{F}$ has poly size, there is a fixed $k$ such that, for large enough $n$,[19]

$$2^{b(n)} \leq n^k$$
$$\Rightarrow b(n) \leq k\lceil \log n \rceil.$$

Define $b'(n) = k\lceil \log n \rceil$ and $\mathsf{bsize}(n) = 2^{b'(n)}$. $\mathsf{bsize}(n)$ is both a power of 2 and an upper bound on $2^{b(n)}$; what remains to be shown is that it can be computed in time $O(\log n)$. We can first compute $\lceil \log n \rceil$ in time $O(\log n)$ by finding the greatest nonzero index of $n$. Next, we can compute $b'(n) = k \cdot \lceil \log n \rceil$ in time $O(\log \log n)$ since $k$ is fixed size and $\lceil \log n \rceil$ has size at most $O(\log \log n)$ (Brent & Zimmermann, 2010). Finally, we compute $\mathsf{bsize}(n) = 2^{b'(n)}$ by simply left-shifting 1 at most $O(\log n)$ times. $\qquad\square$

---

[19]We can compute $\mathsf{bsize}(n)$ for small $n$ using finite lookup.

## B.4 Circuit Families Can Be Padded to Log-Time Size Upper Bounds

Recall that the last $p$ bits of our circuits represent the circuit's output (cf. Section 5.1). In Lemma 4, we consider $\mathcal{F}(x) = \mathcal{F}'(x)$ if and only if the last $p$ bits of $\mathcal{F}$ and $\mathcal{F}'$ agree for all $x$.

**Lemma 4** (Proof in Appendix B.4). *If $\mathcal{F}$ is a log-uniform $\mathsf{TC}^0$ family and $\mathsf{size}_{\mathcal{F}}(n) \leq \mathsf{bsize}(n)$, there exists a log-uniform $\mathsf{TC}^0$ family $\mathcal{F}'$ s.t. $\mathcal{F}(x) = \mathcal{F}'(x)$ for all $x$ and $\mathsf{size}_{\mathcal{F}'}(n) = \mathsf{bsize}(n)$.*

*Proof.* The high level idea is that we can pad $\mathcal{F}$ to a circuit $\mathcal{F}'$ that has size $\mathsf{bsize}(n)$ and simply copies over the $p$ output bits of $\mathcal{F}$ to its own last $p$ bits using identity gates.

We first set $\mathsf{node}_{\mathcal{F}'}$ to copy over the existing circuit and append identity nodes. Let Id denote an identity node. Then $\mathsf{node}_{\mathcal{F}'}$ is defined as:

$$\mathsf{node}_{\mathcal{F}'}(n, i) = \begin{cases} \mathsf{node}_{\mathcal{F}}(n, i) & \text{if } \mathsf{node}_{\mathcal{F}}(n, i) \neq \emptyset \\ \mathrm{Id} & \text{if } \mathsf{node}_{\mathcal{F}}(n, i) = \emptyset \wedge i < \mathsf{bsize}(n) \\ \emptyset & \text{otherwise.} \end{cases}$$

We see that the size of $\mathcal{F}'$ will thus be of size $\mathsf{bsize}(n)$.

Next, we extend $\mathsf{edge}_{\mathcal{F}'}(n, i, j)$ to route the original output bits to the new output bits. Recall that an edge value of 0 means $i$ is the first argument of gate $j$, and an edge value of $-1$ means there is no edge $i \rightarrow j$. Let $k_j = p(n) - (\mathsf{bsize}(n) - j)$ be the index of node $j$ as an output gate in $\mathcal{F}'$. For example, $k = 0$ for the first output bit. Now let $\mathsf{output}_{\mathcal{F}}(n, i, k)$ represent whether node $i$ is the $k$-th output of $F_n$. We can compute $\mathsf{output}_{\mathcal{F}}(n, i, k)$ in terms of $\mathsf{node}_{\mathcal{F}}$ as follows:

$$\mathsf{output}_{\mathcal{F}}(n, i, k) \iff \mathsf{node}_{\mathcal{F}}(n, i + p(n) - k - 1) \neq \emptyset \wedge \mathsf{node}_{\mathcal{F}}(n, i + p(n) - k) = \emptyset.$$

Then $\mathsf{edge}_{\mathcal{F}'}$ is defined:

$$\mathsf{edge}_{\mathcal{F}'}(n, i, j) = \begin{cases} \mathsf{edge}_{\mathcal{F}}(n, i, j) & \text{if } \mathsf{edge}_{\mathcal{F}}(n, i, j) \neq -1 \\ 0 & \text{if } \mathsf{output}_{\mathcal{F}}(n, i, k_j) \\ -1 & \text{otherwise.} \end{cases}$$

The first condition simply copies over the original edges. The second condition adds $p(n)$ new edges (for the different values of $k$) that route the final $p(n)$ nodes of $\mathcal{F}$ to the final $p(n)$ nodes of $\mathcal{F}'$, guaranteeing that the two circuits will compute the same function.

Because both $\mathsf{node}_{\mathcal{F}'}$ and $\mathsf{edge}_{\mathcal{F}'}$ just rely on addition, conditional branching, and a finite number of calls to functions computable in time $O(\log n)$, they are both computable in time $O(\log n)$. $\qquad\square$

## C   Transformer Column Components

In this section, we generally omit layer subscripts for clarity. We assume a pre-norm (Xiong et al., 2020) parameterization of the transformer for concreteness and because this is more standard in newer transformers. However, the results would also hold with the original post-norm (Vaswani et al., 2017).

As mentioned in the main text, we view $\theta_{\mathcal{T}}$ as a concatenation of the parameters for the transformer functions. Thus, if $m$ and $w$ are computable in time $O(\log n)$ and $\theta_{\mathcal{T}}$ is log-uniform, it follows that the parameter vector for each $\phi, s, v, f$, and $\kappa$ is itself log-uniform because we can map indices in the smaller parameter vectors to indices in $\theta_{\mathcal{T}}$ in time $O(\log n)$.

### C.1   Transformer Embeddings

For each position $1 \leq i \leq n$, the transformer embedding function represents token $\sigma_i \in \Sigma$ and its position $i$ with a vector. Let $\mathbf{V}$ be an embedding matrix of size $|\Sigma| \times m$ where each row represents the embedding for some $\sigma$. Let $f : \mathbb{N} \rightarrow \mathbb{D}_p^m$ be computable in time $O(\log n)$. Then,

$$\phi(\sigma_i, i) = \mathbf{v}_{\sigma_i} + f(i).$$

**Lemma 6.** *If $\theta_{\mathcal{T}}$ is log-uniform, then $\phi$ is computable in log-uniform $\mathsf{TC}^0$.*

*Proof.* The embedding block can be expressed as a constant-size computation graph that constructs $\mathbf{V}$, computes $\mathbf{v}_{\sigma_i}$ using an affine transformation, computes $f(i)$, and then, finally, sums $\mathbf{v}_{\sigma_i}$ and $f(i)$. The first step is computable by a log-uniform constant-depth, poly-size threshold circuit family since $\theta_{\mathcal{T}}$ is log-uniform. We can compute an affine transformation via a log-uniform constant-depth poly-size threshold circuit family via Lemma 10. $f(i)$ can be directly computed by the Turing machine constructing the circuit by construction. The sum of the two terms can then be computed by a log-uniform constant-depth threshold circuit of size polynomial in $m$, which is also polynomial in $n$. Since we have a computation graph where all node types are computable by log-uniform, constant-depth, poly-size threshold circuit families, we conclude by Corollary 3.2 that $\phi$ can also be computed by log-uniform, constant-depth, poly-size threshold circuit family. $\square$

## C.2 Self Attention

The two components of the self attention block are $s$, the similarity function, and $v$, the value function. Let $\mathbf{h}_i$ be the hidden state at the previous layer and $\bar{\mathbf{h}}_i = \mathrm{lnorm}(\mathbf{h}_i)$. Then, the similarity function first computes queries and keys, and then takes the scaled dot-product between them:

$$\mathbf{q}_i = \mathbf{W}_q\bar{\mathbf{h}}_i + \mathbf{b}_q$$
$$\mathbf{k}_i = \mathbf{W}_k\bar{\mathbf{h}}_i + \mathbf{b}_k$$
$$s(\mathbf{h}_i, \mathbf{h}_j) = \exp\left(\frac{\mathbf{q}_i^\top \mathbf{k}_i}{\sqrt{m/h}}\right).$$

Then the value function is defined $v(\mathbf{h}_i) = \mathbf{W}_h\bar{\mathbf{h}}_i + \mathbf{b}_h$. We first show that the value function (and also the keys and queries by symmetry) is computable in log-uniform $\mathsf{TC}^0$:

**Lemma 7.** *If $\theta_{\mathcal{T}}$ is log-uniform, then the self-attention component is computable in log-uniform $\mathsf{TC}^0$.*

*Proof.* $v$ is a composition of constructing the parameters (in log-uniform $\mathsf{TC}^0$ since $\theta_{\mathcal{T}}$ is log-uniform), layer norm (in log-uniform $\mathsf{TC}^0$ by Lemma 11), and an affine transformation (in log-uniform $\mathsf{TC}^0$ by Lemma 10). Thus, $v$ is computable in log-uniform $\mathsf{TC}^0$.

Computing $s$ is a constant-depth computation graph. First, we compute $\mathbf{q}_i$ and $\mathbf{k}_i$ and then multiply them, and all of these steps are in log-uniform $\mathsf{TC}^0$. Next, we can compute $m$ and $h$ in time $\mathrm{O}(\log n)$ and build a log-uniform $\mathsf{TC}^0$ circuit that divides the product of the last step by $\sqrt{m/h}$. Finally, we compute $p$-precision $\exp$, which can be expressed in log-uniform $\mathsf{TC}^0$ as multiplication followed by left-shifting. Thus, by Corollary 3.2, $s$ can be computed in log-uniform $\mathsf{TC}^0$.

$s$ and $v$ are log-uniform, so their size $p$ is at most $\mathrm{poly}(n)$. Computing self attention reduces to binary multiplication and division over $\mathbb{D}_p$, and performing iterated addition (summation) over $n$ numbers in $\mathbb{D}_p$. Binary multiplication, binary division (Hesse, 2001), and iterated addition (Merrill & Sabharwal, 2023) can all be computed in log-uniform $\mathsf{TC}^0$, i.e., by a log-uniform, constant-depth threshold circuit family of size at most $\mathrm{poly}(p) \leq \mathrm{poly}(n)$. Thus, self attention can also be computed in log-uniform $\mathsf{TC}^0$. $\square$

## C.3 Activation Block

The activation function $f$ encapsulates the aggregation of the attention head outputs and the feedforward subnetwork of the transformer. $f$ takes as input attention head outputs $\mathbf{a}_{i,1}, \ldots, \mathbf{a}_{i,h} \in \mathbb{D}_p^{m/h}$ and the previous layer value $\mathbf{h}_i$.

The first part of the activation block simulates the pooling part of the self-attention sublayer. The head outputs are first concatenated to form a vector $\mathbf{a}_i$, which is then passed through an affine transformation $(\mathbf{W}_o, \mathbf{b}_o) : \mathbb{D}_p^m \to \mathbb{D}_p^m$ followed by residual connections to form the sublayer output $\mathbf{o}_i \in \mathbb{D}_p^m$:

$$\mathbf{o}_i = \mathbf{W}_o\mathbf{a}_i + \mathbf{b}_o + \mathbf{h}_i.$$

The second part of the activation block first applies layer-norm and then simulates the feedforward subnetwork to compute the next layer vector $\mathbf{h}_i'$. Let $\bar{\mathbf{o}}_i = \mathrm{lnorm}(\mathbf{o}_i)$. Let $\sigma$ be a nonlinearity computable in linear time on its input (in the most standard transformer, ReLU). Then, for affine

transformations $(\mathbf{W}_1, \mathbf{b}_1) : \mathbb{D}_p^m \to \mathbb{D}_p^w$ and $(\mathbf{W}_2, \mathbf{b}_2) : \mathbb{D}_p^w \to \mathbb{D}_p^m$, the feedforward subnetwork can be defined:

$$\mathbf{h}_i' = \mathbf{W}_2 \sigma(\mathbf{W}_1 \bar{\mathbf{o}}_i + \mathbf{b}_1) + \mathbf{b}_2 + \mathbf{o}_i.$$

**Lemma 8.** *If $\theta_\mathcal{T}$ is log-uniform, then $f$ is computable in log-uniform $\mathsf{TC}^0$.*

*Proof.* The activation block can be expressed as a constant-size computation graph where the nodes construct affine transformation parameters, apply affine transformations, compute layer-norm, and compute elementwise nonlinearities. Since each of these nodes is computable by a log-uniform, constant-depth, poly-size threshold circuit family, the activation block is as well. □

### C.4 Output Classifier Head

We assume the output from the transformer is computed as follows. First, $\bar{\mathbf{h}}_1 = \mathrm{lnorm}(\mathbf{h}_1)$. Then, we use a parameter vector $\mathbf{w} \in \mathbb{D}_p^m$ and bias term $b$ to compute:

$$\kappa(\mathbf{h}_1) = \mathrm{sgn}(\mathbf{w}^\top \bar{\mathbf{h}}_1 + b).$$

**Lemma 9.** *If $\theta_\mathcal{T}$ is log-uniform, then $\kappa$ is computable in log-uniform $\mathsf{TC}^0$.*

*Proof.* We can express computing $\kappa$ as a composition of constructing the parameters $\mathbf{w}, b$ and computing the affine transformation. Both parts of this composition are computable by a log-uniform, constant-depth, poly-size threshold circuit family, so computing $\kappa$ is as well. □

## D Neural Net Building Blocks

In this section we analyze the uniformity of common neural net building blocks that are used within the various high-level transformer components.

### D.1 Affine Transformations

Affine transformations are a core part of neural networks used in various parts of the transformer. An affine transformation takes as input parameters $(\mathbf{W}, \mathbf{b}) : \mathbb{D}_p^a \to \mathbb{D}_p^b$ and a vector $\mathbf{x} \in \mathbb{D}_p^a$ and returns $\mathbf{W}\mathbf{x} + \mathbf{b}$.

**Lemma 10.** *For $p = \mathrm{O}(\log n)$, any $p$-precision affine transformation where $\mathbf{W}, \mathbf{b}$ are log-uniform is computable by a log-uniform, constant-size threshold circuit family of size polynomial in $a$ and $b$.*

*Proof.* We first use the uniformity of $\mathbf{W}, \mathbf{b}$ to construct them in $\mathrm{O}(\log n)$ time. For the transformation $\mathbf{W}\mathbf{x} + \mathbf{b}$, first compute each $\mathbf{w}_i \odot \mathbf{x}$ in parallel, where $\odot$ represents elementwise multiplication. Since binary multiplication over polynomial-size numbers is in log-uniform $\mathsf{TC}^0$, this can be done in parallel with log-uniform $\mathsf{TC}^0$ circuits. We then use $b$ log-uniform, constant-depth, poly-size threshold circuit families, each corresponding to an output index, that compute the sum over the $a$ entries of each $\mathbf{w}_i \odot \mathbf{x}$. The affine transformation corresponds to the composition of these two steps, and is thus computable by a log-uniform $\mathsf{TC}^0$ circuit family. □

### D.2 Layer Norm

The layer norm is applied between sublayers in the transformer. Let $\mu = (1/d) \sum_{i=1}^d x_i$. The layer norm $\mathbf{y} \in \mathbb{D}_p^m$ of a vector $\mathbf{x} \in \mathbb{D}_p^m$ is computed, for scalars $a, b \in \mathbb{D}_p$,

$$\mathbf{y} = a \left( \frac{\mathbf{x} - \mu}{\|\mathbf{x} - \mu\|} \right) + b.$$

**Lemma 11.** *If $a, b$ are log-uniform, the layer norm over a vector of size $m$ can be computed by a log-uniform threshold circuit family of constant depth and size polynomial in $m$.*

*Proof.* First compute $m$ using summation over the constant term 1 from 1 to $m$. This summation can be computed by a log-uniform constant-depth threshold circuit family of size polynomial in $m$. Then compute the sum over $\mathbf{x}$ using a similar circuit, and divide them to get $\mu$, using the fact that integer division is in log-uniform $\mathsf{TC}^0$ (Hesse, 2001). We can then compute $\mathbf{x} - \mu$ in log-uniform $\mathsf{TC}^0$.

At this point, we can compute $\|\mathbf{x} - \mu\|$ in log-uniform $\mathsf{TC}^0$ (Hunter et al., 2010), then divide each $\mathbf{x} - \mu$ by the norm in log-uniform $\mathsf{TC}^0$, and then apply the final affine transformation in log-uniform $\mathsf{TC}^0$ (Lemma 10). Thus, computing layer norm is in log-uniform $\mathsf{TC}^0$. $\qquad\square$

## E Arithmetic Complexity

**Lemma 12.** *Given an $m$-bit integer $a$ and $n$-bit integer $b$, we can compute the quotient $\lfloor a/b \rfloor$ and remainder $a \bmod b$ in time $\mathrm{O}(mn)$.*

*Proof.* Let $D(m,n)$ and $M(m,n)$ denote, respectively, the time complexity of dividing and multiplying an $m$-bit integer by an $n$-bit integer. Brent & Zimmermann (2010) give the following fact: $D(m+n,n) \leq \mathrm{O}(M(m,n))$. With the goal of analyzing $D(m,n)$, we apply this as follows:

$$\begin{aligned} D(m,n) &\leq D(m+n,n) \\ &\leq \mathrm{O}(M(m,n)) \\ &\leq \mathrm{O}(mn). \end{aligned}$$

$\square$

Applying Lemma 12 when $a$ has size $\mathrm{O}(\log n)$ and $b$ has size $\mathrm{O}(1)$ says that we can do division in time $\mathrm{O}(\log n)$.