# OpenReview forum: "A Logic for Expressing Log-Precision Transformers"
_NeurIPS.cc/2023/Conference — NeurIPS 2023 poster_

### Official Review · Reviewer_ihd9 · 2023-07-05

**Soundness:** 3 good
**Presentation:** 3 good
**Contribution:** 3 good
**Rating:** 7
**Confidence:** 4

**Summary:**

In this submission, the authors generalize an expressivity result of transformers from Chiang et al. (2023), who showed that finite-precision transformers can be equivalently expressed in a counting generalization of first-order logic. In the paper at hand, the authors generalized this result that log-precision transformers, i.e., that scale with the input size, are expressible in First-order logic with majority votes, FO(M). This formalism is needed to express uniform attention patterns, to, for example, express counting languages such as a^m b^m. The authors provide a well-written introduction to the topic and rigorous proof.

**Strengths:**

- A novel bound for the expressive power of transformers, which generalizes previous results; the problem is interesting to the NeurIPS community. Transformers are ubiquitous.
- The paper is well written; the transformer formalization and FO(M) is well-explained with the NeurIPS community in mind. Examples are provided (section 2).
- Intuition paragraphs are provided after the main results and parts of the proof (second half of the paper), making it ok to follow.

**Weaknesses:**

- Although not necessary for this type of theoretical work, there would have been room to conduct some interesting experimental results to visualize and empirically analyze the authors claims
- The most important lemma (I feel), Lemma 2, is moved to the appendix and scattered across multiple sections there with little intuition given. The reviewer found this the most interesting part and more important for understanding the proof than section 6.1, as the simulation is straight forward.


**Questions:**

- I wonder how the claim that uniform attention is necessary relates to the findings of “ALiBi” of Press et al. (ICLR, 2022)?
- What are the author’s opinions on the next steps to tighten the bound further?
- Would you consider moving Lemma 2 to the main part of the paper? What are your opinions on that?

**Limitations:**

The bound is not tight. The missing tightness of the provided bound would have deserved its own section or a more involved explanation. It is only partially touched on at the end of the conclusion.

---

> ### Author Rebuttal · Authors · 2023-08-09
>
> Thank you for your review and questions!
>
> > Would you consider moving Lemma 2 to the main part of the paper? What are your opinions on that?
>
> We appreciate this point and agree that having some discussion of the Lemma 2 proof in the main paper could give more intuition about why the result at large goes through. The reason we originally put Lemma 2 in the appendix is because its full proof is really quite large: there is a separate sub-lemma for each transformer component in Section C. We thus don’t think we can fit the full proof in the main paper. Perhaps a good middle ground would be to add a paragraph giving a proof sketch of Lemma 2 in the main paper, and then refer the reader to the appendix for the details about each component.
>
> > What are the author’s opinions on the next steps to tighten the bound further?
>
> We believe a promising direction would be to analyze the width of the circuit required to simulate a transformer. Intuitively at least, we believe there should be a connection between the circuit  width and the quantifier depth of an equivalent FO(M) formula. Working out the details here could allow us to say something like “any transformer can be simulated by an FO(M) formula of quantifier depth at most d” (for some constant d).
>
> At the same time, it could be interesting to conduct some experiments on the ability of transformers to learn different formal languages definable in FO(M). Varying crucial properties like the quantifier depth could generate hypotheses about which problems within FO(M) may be out of reach for transformers.
>
> > I wonder how the claim that uniform attention is necessary relates to the findings of “ALiBi” of Press et al. (ICLR, 2022)?
>
> Good question! Under ALiBi, the positional bias in principle will introduce some deviation from uniform attention, although we believe the deviation can be quite small. This is because the parameter norm of the network will likely grow over time (cf. [Merrill et al., 2021](https://arxiv.org/abs/2010.09697)) and the magnitude of qk will scale roughly quadratically in the parameter norm. It follows that the magnitude of qk can dominate the positional bias term of AliBi, especially for the heads with a low value of m (cf. AliBi paper, Section 3). We thus think it is possible to closely approximate uniform attention with AliBi, even if there will be some slight deviation from it.

---

> > ### Comment · Reviewer_ihd9 · 2023-08-18
> >
> > Interesting. Thanks for your reply! I have no more questions.

---

### Official Review · Reviewer_NeeV · 2023-07-06

**Soundness:** 4 excellent
**Presentation:** 3 good
**Contribution:** 3 good
**Rating:** 7
**Confidence:** 4

**Summary:**

This paper provides two contributions which extend previous work showing that finite-precision transformers may be expressed in FOL.
* Firstly, the authors prove that finite-precision transformers can only uniformly attend to bounded-length windows over their input sequence, with the constraints on previous theoretical work (those of Chiang et al. on finite-precision models) being too tight to model the capabilites of real models (which are capable of expressing sentences of the form $a^mb^mc^m$). This motivates them to consider log-precision transformers which are capable of attending to unbounded ranges at $O(\log n)$ precision.
* Secondly, the authors prove that the computation performed by a log precision transformer can be expressed in FO(M) logic, a variant of first order logic presupposing the typical quantifiers (universal, existential) as well as a majority quantifier (if a variable is present $\geq \frac{1}{2}$ of the entries in a set). They prove this by showing that log-precision transformers belong to computation graph families which are computable in log-uniform threshold circuits, which in-turn (leveraging existing results) are expressable in FO(M).

Alongside the proofs for the second claim, they also provide an algorithm which demonstrates how (in principle) one can express computational graph families with circuit families, by breaking up each node in the computational graph into contiguous blocks of circuit gates; also considering the case of log-uniform computation graphs.

**Strengths:**

The paper appears technically sound, though proofs outside of the main text (those in the appendices) were not thoroughly checked (the first claim, regarding limited expressivity of finite-precision transformers, is small and was checked).

The approach taken to demonstrate the FO(M) expressibility draws on existing results from various domains and uses these to extend previous findings on logical expressivity of transformers, thus it is both quite novel and valuable. Whilst these findings remain far from useful for practical mechanistic interpretability, they serve as a useful foundation for work hoping to explore any such applications.

**Weaknesses:**

No major weaknesses were identified.

Two minor weaknesses are:
1. The limited applicability which is only partially addressed in the section on "Mechanistic Interpretability". In particular, the size of a FO(M) sentence required to express a practical transformer would presumably be significant enough that its interpretability would be questionable at best. The authors do conjecture that quantifier depth "of at most 2" may suffice to express transformers, but this has yet to be shown. This criticism is applicable to other technical attempts to produce discrete / tree-like representations of transformers, and so is not a significant drawback, besides saying that any attempts to motivate such research from a practical perspective seem highly optimistic at present.
2. The paper is quite dense in parts, which is understandable. However, two potentially useful figures could be envisioned: 1) a high-level overview of the proof (log-precision transformers -> computational graph -> TC -> FO(M)) 2) A figure illustrating the contiguous circuit construction algorithm from $6.1

## Nitpicks
* 55: "powerful enough *to* express"
* Example 5: Seems incomplete?
* 252: "the second of which" should probably read "one of which" as it is otherwise somewhat confusing why "the second" is being specified
* 310: "without loss of generality" -> "w.l.o.g" to be consistent with previous use
* 355: "Challenging the premise of a rigid division between symbolic and neural models". The mathematical sense in which this is being challenged is quite loose, and so it is not clear that this meaningfully "challenges" the division in the way it is typically framed (in a sense, matrices are symbolic, especially if a model is quantized, but this does not make the model *symbolic* in the way a logician would find useful)


**Questions:**

None

**Limitations:**

Theoretical limits of the proofs and claims are precise and well-stated. Claims around practical applicability of these results may be slightly overstated.

---

> ### Author Rebuttal · Authors · 2023-08-09
>
> Thank you for your review! We appreciate your suggestions for making the paper less dense. If accepted, we will add a high-level overview of the proof toward the end of Section 4, as you suggest. A figure illustrating the Algorithm from 6.1 would also be nice if space permits - we will consider whether it is possible to add this without pushing too much other important content to the appendix.
>
> Regarding the potential interpretability limitations, we agree that a very large/deep logical formula on its own wouldn’t help anyone better understand a transformer (although it could enable finding meaningful substructures or facilitate formal verification). Therefore, we think that the quantifier depth of FOM formula is quite important for its interpretability, and are actively thinking about a depth-2 / low-depth FOM simulation as a follow-up.  (see also response to R4)

---

> > ### Comment · Reviewer_NeeV · 2023-08-14
> >
> > Thank you for your response.
> >
> > A high-level overview will indeed be appreciated, and the omission of a figure given the space-constraints and justifiable inclusion of all current main-paper contents is reasonable.
> >
> > The limitations to interpretability are not a strong limitation of this work, but I am pleased to hear that you are already considering bounds on the sizes of circuits!
> >
> > I retain the previous score and recommend this paper for acceptance as sound theoretical work which provides novel insights and lays the groundwork for more practical interpretability-focused work going forward.

---

### Official Review · Reviewer_kdtb · 2023-07-06

**Soundness:** 4 excellent
**Presentation:** 3 good
**Contribution:** 3 good
**Rating:** 7
**Confidence:** 3

**Summary:**

The paper presents a theoretical analysis on the expressiveness of transformer-based models. In particular, the authors have managed to prove that the any log-precision transformer is equivalently expressible as normal first-order logic plus majority-vote quantifiers, FO(M). This yields the tightest known upper bound of log-precision transformers.

**Strengths:**

The result is of significant theoretical interest and can potentially bridge well-studied first-order logic and widely used transformer architecture (e.g., via mechanistic interpretability work). The paper is nicely structured and with well crafted examples. To the best of my knowledge, I don't see major gaps in their proof derivations.

**Weaknesses:**

I don't see any significant weakness in this paper.

**Questions:**

Minor:
- Figure 1 on page 1 is admittedly a great motivation example but we don't know the exact meaning of an and b until page 4. It is perhaps a good idea to shed more words about Figure 1 earlier.
- line 80, 'atend'

**Limitations:**

The authors have adequately addressed the limitations.

---

> ### Author Rebuttal · Authors · 2023-08-09
>
> Thanks for your review! We appreciate that you found our work to be of theoretical interest. As you suggest, we will clarify the notation used in Figure 1 in the caption so that it is more self-contained.

---

### Official Review · Reviewer_27Gc · 2023-07-07

**Soundness:** 3 good
**Presentation:** 3 good
**Contribution:** 3 good
**Rating:** 7
**Confidence:** 3

**Summary:**

In this paper, the computational power of transformers is studied subject to floating point precision and related to that of first-order logic. In particular, the authors show that (1) fixed-precision is not sufficient to compute uniform attention for arbitrary context lengths and (2) log-precision can be simulated by first-order logic with majority. The latter is shown by establishing that log-precision transformers can be simulated by a family of log-uniform circuits which is known to be equivalent to first-order logic with majority.

**Strengths:**

- Some of the existing theoretical results on the computational power of transformers seem to only partially support observations in practice. For example, it has been shown that the language $a^nb^n$ provably can not be recognized by a fixed-precision transformer while experiments indicate the opposite. The paper contributes to better understanding this gap by deriving theoretical results for log-precision transformers that seem to resemble observations in practice although practical implementations are in fact fixed-precision.
- While the expressive power of log-precision transformers has been characterized by first-order logics in previous work, the paper established the tightest bound so far.
- The extensive use of transformers and language models in formal domains such as programming languages and theorem proving makes the connection between transformers and formal logics particularly interesting.


**Weaknesses:**

- The role of positional encoding is not discussed in the analysis. Yet, fixed-precision and log-precision has direct implications on the ability to represent a positional encoding.
- Merrill & Sabharwal (2023) show that a log-precision transformer can be simulated by a uniform $TC^0$ circuit family which at first seems fairly similar to the result in this paper. Merrill & Sabharwal bound space and prove the result for logspace-uniform $TC^0$ whereas in this paper the authors bound time and prove the result for log-uniform $TC^0$. I think this is not obvious and making it more explicit would help to understand the contribution of this paper.
- No definition of circuit complexity is given in the paper.


**Questions:**

- Is there a straightforward argument that generalizes the results for decoder architectures or are the results in fact limited to encoder architectures?
- In the conclusion the threshold circuit family is referred to as highly uniform. What is the meaning of highly in this context?


**Limitations:**

- The definition of the transformer resembles the encoder part of the full transformer architecture. If the results do not generalize to the transformer decoder this should be addressed as a limitation.

---

> ### Author Rebuttal · Authors · 2023-08-09
>
> Thank you for your review! We’d like to address some of your comments and suggestions around the role of positional encodings, different notions of uniformity, and other aspects of the paper.
>
> ## Positional encodings
>
> > The role of positional encoding is not discussed in the analysis. Yet, fixed-precision and log-precision has direct implications on the ability to represent a positional encoding.
>
> As discussed in C.1, our results go through for any positional embeddings that are log-precision and computable in time O(log n) (where n is seq length) as a function of the natural number i representation position. Because it takes O(log n) bits to uniquely specify a natural number in the range [0, n], these assumptions are met for any natural type of positional encoding. We will better highlight the role of positional encodings in Section 4.
>
> ## Logtime vs. logspace uniformity
>
> > Merrill & Sabharwal (2023) show that a log-precision transformer can be simulated by a uniform circuit family which at first seems fairly similar to the result in this paper. Merrill & Sabharwal bound space and prove the result for logspace-uniform whereas in this paper the authors bound time and prove the result for log-uniform. I think this is not obvious and making it more explicit would help to understand the contribution of this paper.
>
> Thanks for the feedback! We will clarify the distinction when we introduce logtime-uniformity in the Preliminaries.
>
> Relatedly, you asked what we mean by “highly uniform” in the conclusion. We are simply referring to logtime-uniform as opposed to logspace-uniform, which we will clarify. We say “highly uniform” because logtime uniformity is so strong that it collapses circuit families to logical formulas, meaning there is just a single description of the computation for any input size.
>
> ## Other questions
>
> We will add some high-level background on circuit complexity as well as references to standard texts at the beginning of the preliminaries section.
>
> > Is there a straightforward argument that generalizes the results for decoder architectures or are the results in fact limited to encoder architectures?
>
> This is a good question that we have also been thinking about recently. There is not a simple result or corollary that holds for decoder-models, but it is possible to use these results as a tool to obtain a different upper bound for decoder architectures. As this is quite involved in its own right it is out of scope for this paper, but it forms the basis of our current follow-up research.

---

> > ### Comment · Reviewer_27Gc · 2023-08-17
> >
> > Thank you for your response. The proposed changes address all weaknesses sufficiently, and I will raise my score accordingly.
> >
> > Regarding the decoder architecture, I do not consider it a weakness that it is not further discussed in the paper, but I urge the authors to clarify in the paper that the results apply only to encoder architectures.

---

### Decision · Program_Chairs · 2023-09-21

**Decision:**

Accept (poster)

**Comment:**

The reviews are unanimous in favor of acceptance. First of all, this is an important question. Since the take-off of large language models, whether and how they can reason becomes sort of a mystery. Some prior work focuses on the revelation of some of their properties through empirical investigation. Yet, principled theoretical analysis is relatively scarce. The conclusion of this paper is interesting that LLMS describe logical rules and its inference is equivalent to resolution over those logical rules. The simplification made in this paper is reasonable. We have decided to recommend this one for poster presentation. Yet, after discussion we believe still there is room for improvements. More discussion regarding related work would present readers with a more comprehensive understanding about how to understand this paper’s insight against the backdrop of other theoretical work. For example, now some paper thinks LLMs are similar to Turing machines, and some papers thinks LLMs construct “templates” for reasoning implicitly, which are similar to logical rules. Maybe the discussion would go over the page limit. But considering it in the appendix is a plus. Secondly, we would like this paper to have a more comprehensive discussion in the future work. There is still some aspects in the LLMs that are left out in this paper. For example, the prompting mechanism. Can we extend the results here? Of course, the suggestions regarding this paper’s notation and presentation is also hugely recommended to be incorporated.